# The Determinant of Tau Spreading in Alzheimer’s Disease: Dependent on Senile Plaque, Neural Circuits, or Spatial Proximity?

**DOI:** 10.3390/ijms262412088

**Published:** 2025-12-16

**Authors:** Yuichi Riku, Jean-Pierre Brion, Kunie Ando, Toshiki Uchihara, Yasushi Iwasaki

**Affiliations:** 1Institute for Medical Science of Aging, Aichi Medical University, Nagakute 480-1195, Aichi, Japan; 2Department of Neurology, Nagoya University, Nagoya 466-8550, Aichi, Japan; 3Alzheimer and Other Tauopathies Research Group, ULB Center for Diabetes Research (UCDR), Faculty of Medicine, ULB Neuroscience Institute (UNI), Université Libre de Bruxelles, 1050 Bruxelles, Belgium; 4Okinawa Chubu Hospital, Uruma 904-2293, Okinawa, Japan

**Keywords:** Alzheimer’s disease, tau, neurofibrillary tangle, senile plaque, Aβ, neuropathology, autopsy, amyloid cascade, tauopathy, protein propagation

## Abstract

Alzheimer’s disease (AD) is neuropathologically characterized by tau-immunopositive neurofibrillary tangles (NFTs) and amyloid-β (Aβ)-immunopositive senile plaques. According to the widely accepted amyloid cascade hypothesis, Aβ pathology represents the upstream event in AD pathophysiology and induces tau aggregation. However, numerous studies have suggested that tau aggregates correlate more closely with neuronal loss and regional brain atrophy than with Aβ depositions. Tau aggregation in AD demonstrates a hierarchical spreading pattern beginning in the transentorhinal cortex, but the mechanisms underlying this spreading manner of lesions remain to be elucidated. This review aims to address current controversies regarding tau pathology in AD from the perspectives of both the ‘amyloid cascade’ and ‘tauopathy’ hypotheses. From the ‘amyloid cascade’ viewpoint, Aβ deposition prominently involves distal axon and axon terminals, and in some regions, there are anatomical correspondences between axonal Aβ pathology and cytoplasmic tau aggregations (e.g., a close relationship between senile plaques in the molecular layer of the hippocampal dentate gyrus and NFTs in the transentorhinal cortex). Nevertheless, this model cannot explain the whole body of hierarchical spreading of tau aggregation because notable spaciotemporal discrepancies also exist in many regions. From the ‘tauopathy’ perspective, the distribution of tau aggregates in AD involves key nodes within the memory circuits. Also, experimental studies have suggested that patient-derived tau exhibits seeding and neuron-to-neuron propagation properties. Interestingly, tau aggregation in AD appears to spread laterally in a proximity-dependent, cortico-cortical fashion rather than along long-range memory circuits. This contrasts with the system-selective, poly-nodal degenerations seen in four-repeat tauopathies, amyotrophic lateral sclerosis, or spinocerebellar degenerations. Moreover, the proportions of three-repeat and four-repeat isoforms shift during the maturation of NFTs in AD. Overall, spreading patterns of tau-pathology in AD cannot be fully explained by Aβ pathology and also differ from the system degeneration seen in other tauopathies.

## 1. Introduction

Alzheimer’s disease (AD) encompasses a clinical and neuropathological entity that corresponds to the syndrome of Alzheimer’s type dementia. The World Health Organization (WHO) estimates that Alzheimer’s disease contributes to 60–70% of dementia cases, and a world-wide surveillance effort revealed an increased prevalence of Alzheimer’s type dementia for the last 30 years even when the ages are standardized [1]. Following clinical studies from the US, Alzheimer’s type dementia and related cognitive impairment occur in 10% or more among the population > 65 years; the burden is estimated at USD 627 billion in aggregate [2]. The social impact of this disease is severe and growing.

In 1907, Alois Alzheimer first reported the clinical and postmortem findings of a 51-year-old female patient who presented with delusional jealousy and progressive dementia, and who was autopsied four and half years after syndrome onset [3]. His work addressed the two essential neuropathological hallmarks of AD: neurofibrillary tangles (NFTs) in the neurons, identified by Bielschowsky silver impregnation, and senile plaques in the neuropil, of which he particularly emphasized the former. Around the same time, neuritic plaques associated with abnormal neurites were clearly described by Oskar Fischer [4]. Although senile plaques had been already described earlier by Blocq and Marinesco and by Redlich, their pathogenic significance remained uncertain [5,6].

Today, NFTs are known to consist of intraneuronal aggregates of hyperphosphorylated tau, while senile plaques are extracellular deposits of amyloid-β (Aβ) that are surrounded by dystrophic neurites. The widely accepted ‘amyloid cascade’ hypothesis positions Aβ as the initiating event of AD-pathogenesis, acting as the upstream driver of tau aggregation and subsequent neurodegeneration [7,8]. Biomarkers that reflect dynamics of these pathologic proteins, including tracing of tau and Aβ with positron emission tomography (PET) and measurement of phosphorylated tau (e.g., p-tau 181, p-tau 217) and Aβ levels in plasma or cerebrospinal fluid [9,10,11], demonstrate reliable correlations with disease progression.

Postmortem studies have clarified that the spreading of tau aggregates follows a stereotyped hierarchical progression, starting in the transentorhinal cortex and advancing through the hippocampus to widespread neocortical regions [12]. Aβ deposits also exhibit a characteristic spreading manner, which is thought to originate in various cortical areas [13]. From the perspective of the amyloid cascade, the spatial distribution of tau aggregates would therefore be expected to follow that of Aβ deposition, based on the anatomical relationships between cytoplasm (tau) and neuritic (Aβ) pathology. Otherwise, a tau-autonomous mechanism may also contribute to the spreading of tau-aggregates, independent of Aβ. Neuronal loss and the spread of tau aggregates preferentially involve specific nodes of memory-related circuits, suggesting that AD may represent a form of ‘system degeneration’ involving memory networks. The selective vulnerability of certain neuroanatomical systems may reflect the propagation of neurotoxic protein aggregates, excitotoxicity associated with system-specific neurotransmitters, or intrinsic cellular vulnerabilities [14,15].

Nevertheless, critical gaps remain in our understanding of AD pathogenesis. In particular, the following two points are unresolved: (1) the nature and cause of the spaciotemporal discrepancies between Aβ and tau pathologies; and (2) whether the spreading manner of tau aggregation really reflects a system degeneration of memory circuits. This review first provides an overview of the neuropathological findings of AD and then discusses the controversies about tau pathology in AD from the perspectives of ‘amyloid cascade’ and ‘tauopathy’.

List of chapters:IntroductionAβ and Tau Pathology in AD2.2Neuropathology of AD2.3Is Tau a Bystander of Aβ in AD Pathology?Controversies of Tau Pathology in AD3.1Spatiotemporal Discrepancy Between Senile Plaque and NFTs in the Human Brain, from the Viewpoint of the ‘Amyloid Cascade’3.2Is AD a System Degeneration of Memory Circuits? From the Viewpoint of ‘Tauopathy’Conclusions

## 2. Aβ and Tau Pathology in AD

### 2.1. Neuropathology of Alzheimer’s Disease

Brains of AD patients exhibit characteristic atrophy of the parahippocampal gyrus (including the entorhinal and transentorhinal areas) and of the hippocampus. In advanced stages, diffuse cortical atrophy becomes evident (Figure 1). Microscopically, AD pathology is characterized by a combination of neurofibrillary tangles (NFTs) and senile plaques. The current diagnostic criteria require the severity of both NFTs and Aβ deposits to be above cutoff, and for Aβ, the presence of neuritic plaques on silver impregnation or thioflavin is mandatory [16]. This combined assessment of NFT and senile plaques arises from the fact that neither NFTs nor senile plaques alone can reliably distinguish AD from normal aging. Although Braak and colleagues established immunohistochemical staging of NFTs on the basis of a large autopsy sample size, their study did not include clinicopathological correlations [17]. Similarly, while the burden of senile plaques correlates with AD pathology, it often lacks a distinct boundary separating AD from age-related changes [18].

NFTs are flame-shaped, filamentous intraneuronal inclusions which are strongly labeled with silver impregnation and thioflavin (Figure 2). NFTs are dense aggregations of hyperphosphorylated tau that are composed of fibrils, ultrastructurally termed paired-helical filaments [19]. Early-stage lesions often exhibit diffuse and amorphous tau immunopositivity in the neuronal cytoplasm but lack a defined tangle structure or argyrophilicity, known as “pre-tangles”. Pre-tangles are considered to represent the early phase of tau aggregation, dominated by tau oligomers rather than mature fibrils [20,21] (Figure 2). Tau also aggregates within the neurites, forming neuropil threads, which can be identified in macroscopic observations of tau-immunostained preparations in severely affected regions (Figure 2).

Tau is encoded by the *microtubule-associated protein tau* (*MAPT*) gene, and alternative splicing of exons 2, 3, and 10 yields six isoforms of tau [22]. These tau isoforms are largely classified into two groups: three-repeat (3R) isoforms lacking exon 10, and four-repeat (4R) isoforms containing exon 10 [23]. Both 3R and 4R isoforms are present in the NFTs of AD; pure 4R aggregates are observed in progressive supranuclear palsy (PSP) and corticobasal degeneration (CBD), whereas pure three-repeat aggregates are typical of Pick body disease [24,25]. The hierarchical distribution of tau pathology progression, as shown by Braak et al., demonstrates initial involvement of the transentorhinal area of the parahippocampal gyrus, and spreads to the entorhinal area and hippocampus (Braak stages I–II), then lateral and superior temporal gyri (stages III–IV), followed by widespread neocortices (stages V–VI) [12,17].

Senile plaque is characterized by the deposition of Aβ in the neuropil, in the form of extracellular deposits that presumably interact closely with the neuronal and glial membranes [26]. Although the classification of senile plaques has been complicated because of methodological differences (e.g., anti-Aβ immunohistochemistry, thioflavin, or silver impregnations), anti-Aβ immunostaining is largely classified into two major types: neuritic plaques and diffuse plaques (Figure 3). Neuritic plaques comprise a dense amyloid core surrounded by a fuzzy halo. Argyrophilic and thioflavin-positive dystrophic neurites form a crown around the amyloid core, and some of them contain tau aggregates. The core and crown can be identified by hematoxylin-eosin (H&E) staining. In contrast, diffuse plaques are fuzzy deposits, lacking distinct core and crown structures. Diffuse plaques are hardly detectable on H&E staining and show weak or no positivity on silver impregnation, although the Campbell–Switzer method recognizes diffuse plaques well. Familial AD associated with *presenilin-1* (*PSEN1*) or *amyloid precursor protein* (*APP*) mutations may also contain cotton wool plaques that are characterized with a clear boundary but lack core/crown structures. Cotton wool plaques are quite exceptional among sporadic cases [27]. APP is cleaved by β-secretase at its N-terminus and by γ-secretase (which includes PSEN1) at its C-terminus, generating several Aβ species of different molecular weights. Aβ_40_ predominates in neuritic plaques, particularly within the amyloid core, whereas Aβ_42_ is enriched in diffuse plaques [28].

There is a hypothesis that diffuse and neuritic plaques are on a pathological continuum, corresponding to disease progression in AD. Neuropathological studies have reported that preclinical or early AD cases are associated with a diffuse plaque-dominant state, whereas advanced AD cases are associated with a neuritic plaque-dominant state [29,30]. Supportively, a basic study using APP-transgenic (tg) mice has clarified that Aβ deposition precedes neuritic swelling, and the neuritic changes were ameliorated after anti-Aβ immunotherapy [31]. However, controversies may exist in the interpretation of plaque subtypes; an exogeneous Aβ-injection into APP-tg models has revealed that the morphological subtypes of Aβ deposits depend on both the host and source of the agent, suggesting the presence of distinct Aβ strains with variable biological activities [32].

It is also known that neuritic plaques often attract glial cells, particularly microglia, whereas diffuse plaques do not [33] (Figure 3). These findings emphasize the importance of neuroinflammation in Aβ-related pathology in AD. Disease-associated microglia may act as sensors of neuronal damage, initiating inflammatory cascades [34]. Nevertheless, the question of whether plaque-associated microglia are merely reactive or actively drive plaque-induced neurodegeneration remains controversial [35].

The severity and distribution of tau pathology are more strongly correlated with brain atrophy and the clinical presentation of AD than Aβ deposition, though Aβ pathology may have synergistic effects with tau-derived neuronal damage [5,32,36,37]. Tau and Aβ PET scan studies have confirmed that tau pathology has a higher correlation than Aβ pathology with cognitive status and its evolution [38]. Basic studies also revealed that a reduction in endogenous tau resulted in rescuing neurodegenerative phenotypes in an *APP*-mutant model, highlighting the pivotal role of tau in AD pathophysiology [39,40].

### 2.2. Is Tau a Bystander of Aβ in AD Pathology?

The concept of the amyloid cascade is widely accepted as an explanation of AD pathogenesis. In brief, it proposes that an increased Aβ_42_/Aβ_40_ ratio and subsequent plaque formation are crucial for AD pathogenesis. In this theory, tau aggregation is considered to be a downstream event of Aβ deposition. Indeed, clinical and neuropathological findings support this hypothesis. Mutations in *PSEN* and *APP* are causative of familial AD, and importantly, these mutations are associated not only with Aβ deposits but also with severe NFT pathology. Actually, the original case described by Alzheimer also had a mutation in *PSEN1* [41]. Moreover, *APP* is located on chromosome 21, and Down syndrome (21 trisomy) is linked to a high prevalence and early onset of AD; both Aβ and tau aggregation are facilitated in this setting [42] (Figure 4). The extent of Aβ deposition is correlated with that of tau aggregates among AD cases and an aged population [13] or in AD model mice with *APP/MAPT*-tg [43]. By contrast, mutations of the *MAPT* gene, encoding tau, are causative of frontotemporal dementia with parkinsonism (FTDP), demonstrating marked tau aggregation but little or no Aβ deposition [44].

However, the direct pathway from Aβ-deposition to tau-aggregation remains largely unknown and has proven difficult to reproduce experimentally. The iatrogenic transmission of Aβ pathology does not induce a tau pathology. It has been reported that patient-derived Aβ has a prion-like property of transmission, whose aggregative seed demonstrates transmission across individuals or species [45]. However, when Aβ is transmitted in vivo, Aβ deposition alone does not seem to cause AD-type tau pathology (3R and 4R mixed tauopathy and formation of NFTs). An example is the human transmission of cerebral amyloid angiopathy, often associated with parenchymal Aβ deposits, through grafts of dura mater, without or with minimal development of a tauopathy, even after decades [46]. Another example is the iatrogenic transmission of Aβ via the injection of cadaver-extracted human growth hormone (crGH). These batches of crGH were contaminated with prions, Aβ, and tau. Postmortem studies of the recipients revealed Aβ deposits, including senile plaques and CAA, in combination with Creutzfeldt–Jakob disease (CJD). Tau aggregates were occasionally detected in these cases, mainly showing dots or neuritic patterns near the plaques and very occasional NFTs; the prevalence, morphology, severity, and distribution of tau pathology differed from and was far smaller than those in AD [47,48,49]. Intracerebral injection of patient-derived Aβ into *APP*-tg mice did not result in tau aggregation [50]. Another study reported that the induction of patient-derived Aβ dimers in primary hippocampal neurons produced neuritic dystrophy and facilitated the phosphorylation of tau within dystrophic neurites, but did not lead to the formation of NFTs [51]. Although injection of Aβ in wild-type mice does not induce the formation of NFT, the injection of Aβ in mutant-*MAPT* Tg mice (which develop NFTs) increases the formation of tau aggregates [52] and crossing *APP* mice with mutant *MAPT*-Tg mice also greatly increases the formation of tau aggregates [53,54]. The injection of human tau seeds also induces more neuritic tau pathology in the presence of amyloid plaque [55,56]. These facts highlight the accelerating role of Aβ in the induction of more NFT, when NFTs are already present.

Neuropathological studies have revealed that NFT can arise in the absence of or with sparse Aβ deposition among older people, termed primary aging-related tauopathy (PART). The morphology and topographic distribution of NFTs in PART do not differ from those in early AD [57]; a subset of PART may be symptomatic, termed senile dementia with NFT (SDNFT) [58]. As mentioned above, *MAPT* mutations cause FTDP; however, certain variants such as V337M and R406W yield AD-type tau filaments containing both 3R- and 4R-tau isoforms even in the absence of Aβ deposition [59,60]. Collectively, these findings indicate that tau aggregation can occur without Aβ deposits. Tau aggregates in AD might be regulated by multiple facilitating or preventive factors [61], even though Aβ deposition is a powerful driver of tau accumulation in AD.

Two questions, however, remain unresolved in this theory. First, it is highly controversial whether PART is really on a continuum of tau aggregation of AD or a distinct entity from AD. It has been described that conformation-dependent tau antibodies distinguish pathological tau in AD from other tauopathies but do not distinguish AD from PART [62], and tau seeding activity from the transentorhinal/entorhinal cortices (where tau pathology starts early) does not differ between AD and PART [63]. PART filaments are identical to those of AD and made of 3R and 4R tau isoforms [64]. However, the opposite findings are also reported; the seeding activity of neocortical tau in PART differs from that of early AD brains containing frequent neuritic plaques [65]. This result indicates that biochemical properties of insoluble tau in NFTs differ between AD and PART in the neocortex. Second, NFTs never progress beyond Braak stage IV in the absence of Aβ [16], arguing against a simple pathological continuum between PART and AD. At least, these findings suggest that Aβ deposition is necessary for the full progression of tau pathology characteristic of definite AD and that PART might play a role in the initiation of AD-related tau pathology, leading to its Aβ-induced progression. However, the precise molecular mechanisms linking Aβ to tau aggregation and the factors modulating this relationship remain incompletely understood.

## 3. Controversies of Tau Pathology in AD

### 3.1. Spatiotemporal Discrepancy Between Senile Plaque and NFTs in the Human Brain, from the Viewpoint of the ‘Amyloid Cascade’

The spaces between the amyloid core and halo contain the crown, composed of swollen neurites, which are usually negative according to Aβ immunostaining. Neuropathological studies have revealed that the swollen neurites involved in neuritic plaques fundamentally express epitopes of axons and axon terminals rather than dendritic epitopes [66,67]. This finding, together with the sparse presentation of senile plaques in white matter, indicates that neuritic plaques prominently involve the distal portion of the axon (Figure 5).

The swollen axons within the neuritic plaques abundantly contain hyperphosphorylated tau aggregates (Figure 5). Studies using multiple dissections of autopsied AD tissues cleared with the CLARITY protocol revealed that intra-axonal tau aggregates of the fornix were continuously extended to the mamillary body and made direct contact with senile plaques in the mammillary body [68,69]. Dystrophic neurites in senile plaques in the neocortex contain tau aggregates when NFTs are present in the same areas but not when NFTs are absent, indicating a close spatial relation between local neurons containing NFTs and plaque-associated tau aggregates [70], although the intra-axonal tau aggregates do not always connect to NFTs [71]. Interestingly, for PART, a recent study suggested an opposite sequence, in which tau aggregation in the entorhinal neurons (pre-α) starts from the dendrites and then spreads to the cytoplasm and axons [72]. Overall, neuritic tau aggregation in the senile plaque is strongly correlated with cytoplasmic NFTs, although it remains unclear whether the former always retrogradely develop the latter.

If we resort to the amyloid cascade hypothesis, it can be expected that NFTs should be enriched in neurons whose distal axons are involved in neuritic plaques. This speculation seems to be partially reasonable because Aβ-vulnerable regions often receive innervation from areas that develop early NFT pathology. For example, the upper molecular layer of the hippocampal dentate gyrus preferentially exhibits Aβ deposition and receives abundant glutamatergic innervation from the entorhinal cortex, one of the early regions affected by NFTs. Correlations between senile plaque burden in the dentate gyrus and NFT burden in the entorhinal cortex have been reported [73,74]. Similarly, early Aβ deposition in the cerebral neocortices corresponds to cholinergic innervation from the basal forebrain, which also develops NFTs during the early phase of AD [75].

However, mismatches between the Aβ and tau pathologies are also noted despite these anatomical correspondences. As discussed in the next chapter, tau aggregation in AD spreads laterally through adjacent cortical areas in a cortico-cortical manner with a smooth gradient, starting from the transentorhinal cortex. It is questionable whether such a highly hierarchical, lateral spreading of tau aggregation arises from distant effects by Aβ deposition, whose spread may start from any cerebral cortices varying among individuals. Moreover, certain regions show discordant pathology: the locus ceruleus is one of the earliest sites of NFT formation and abundantly projects to the cerebellum. Nevertheless, the cerebellum shows Aβ deposition at the latest stage of AD pathology. Similarly, the mamillary bodies and thalamus, which receive abundant innervation from the hippocampus, show relatively sparse Aβ accumulation. A recent study on a large sample size has addressed a mismatch between the Aβ and tau pathologies in subsets of AD cases that display severe Aβ deposition (including extensive neuritic plaques) but unexpectedly sparse NFTs [76].

Overall, Aβ seems to be a powerful facilitator of tau aggregates at a cellular level. However, spatial distribution and temporal spreading of tau aggregates demonstrate some discrepancies from what is expected from Aβ deposits in the context of cytoplasm–axon relationships.

### 3.2. Is AD a System Degeneration of Memory Circuits? From the Viewpoint of ‘Tauopathy’

From the viewpoint of tauopathy, AD can be defined as a form of selective ‘system degeneration’ primarily affecting the memory network [77,78]. This notion arises from the observation that tau aggregation begins in the transentorhinal cortex and extends to the hippocampus and associative neocortices [79], and this spreading course contains some critical nodes of the limbic memory system, including the parahippocampal–hippocampal loop and Papez circuit [80], in AD brains. The selective and highly hierarchical spreading of tau aggregates proposes a tau-autonomous mechanism for tau pathology in AD.

Neuropathological studies of various neurodegenerative disorders have revealed that atrophy, neuronal loss, and accumulation of neuron-aggregative proteins tend to develop along specific neuroanatomical systems that are distantly connected with long tracts beyond the white matter, a process termed ‘system degeneration’. Representative examples include amyotrophic lateral sclerosis (ALS) and frontotemporal lobar degeneration (FTLD). In TAR DNA-binding protein 43kDa (TDP-43)-related ALS (ALS-TDP), upper and lower motor neurons that are distantly connected by the pyramidal tract are affected, showing neuronal loss and TDP-43 aggregation in both areas, resulting in a system degeneration of motor neurons [81,82]. In addition, TDP-43-related FTLD (FTLD-TDP) demonstrates system degeneration along corticostriatal circuits, which are critical for executive functions, mood regulation, and language generation [83,84]. A loss of projection neurons and their axon terminals, combined with TDP-43 aggregates, has been observed across the cerebral cortex, neostriatum, and globus pallidus or substantia nigra [85,86].

In addition to ALS/FTLD, various neurodegenerative disorders including PSP (dentato-rubral or pallido-nigral/luysian degeneration), CBD (cortico-striato-nigral degeneration), and multiple system atrophy (MSA) (olivo/ponto-cerebellar or striato-nigral degeneration) demonstrate system degeneration, showing selective, longitudinal, and poly-nodal impairment along these functional systems [87].

Importantly, the extension pattern of AD-related tau aggregation depends on spatial proximity, with lateral spreading toward adjacent areas resembling the ‘spreading of stain’. When the concept of a ‘system’ is considered, the propagation of tau aggregates in AD tends to terminate in a mono-nodal manner (Figure 6). For example, the Papez circuit, a classical memory system comprising the hippocampus, fornix, mammillary body, anterior nucleus of thalamus, cingulate gyrus, and parahippocampal gyrus, is known to be impaired in AD [80]. Indeed, this system contains AD-vulnerable nodes such as the hippocampus and the parahippocampal gyrus. However, the gradients of tau aggregates and neuronal loss do not necessarily follow the anatomical connection of the circuit; for example, the mammillary body, the next node after the hippocampus, is not preferentially involved in NFTs and neuronal loss [88].

A similar phenomenon is observed in the hippocampal loop [89]. The glutamatergic memory circuit originating from the entorhinal cortex mainly innervates the granule cells of the hippocampal dentate gyrus via the perforant path, followed by CA3 and CA4 via mossy fibers, and finally reaches CA1 via Schaffer collaterals. In this system, the entorhinal cortex is highly vulnerable to tau aggregation, whereas the dentate gyrus and CA3/CA4 are relatively spared; subsequently, CA1 and the subiculum are again vulnerable (Figure 6). Thus, although the memory circuits contain tau-vulnerable regions, neuronal loss and tau aggregation cannot be aligned along a single gradient within these systems. These findings support the view that tau aggregates in AD may spread along short-range neural connections, such as cortico-cortical projections, rather than long tract-dependent circuits [90].

A remarkable supporting finding comes from an autopsied AD patient who had undergone brain surgery 27 years earlier [91]. During surgery, a portion of the frontal cortex was dissected together with a tumor. At autopsy, an island-like cortical area was found to be isolated from the surrounding frontal cortex due to the surgical dissection, remaining connected only through long tracts in the deep white matter. Interestingly, the isolated cortex demonstrated diffuse plaques but not NFTs and neuritic plaques, although these were abundant in other cortical areas. This finding suggests that AD-related tau aggregation preferentially involves cortico-cortical tracts and their associated neurons rather than long-tract projections.

Experimental studies have clarified the propagative properties of tau aggregates derived from patients with AD, PSP, or CBD. Prion-like mechanisms of extracellular release and the propagation of tau aggregates along neuroanatomically connected areas might be central to this process [92,93,94]. Neuronal uptake of induced tau and seed-dependent aggregation have been reported using both in vitro and in vivo models expressing mutant [95,96] or wild-type [97,98] human tau. The hypothesis that tau seeds propagate via short cortico-cortical tracts in the AD brain, resulting in lateral spreading of tau aggregates, is therefore compelling.

However, questions remain regarding tau pathology in AD. As addressed at the beginning of this article, tau aggregates may mature from pretangles to NFTs and ultimately remain as ghost tangles, which lack recognizable cellular structures of the cytoplasm and nucleus. Importantly, it has been reported that tau isoforms change from a 4R-tau-rich state (pretangles and early NFTs) toward a 3R-tau-rich state (mature NFTs and ghost tangles) during the maturation of tau aggregates [24]. Upon a cross-sectional observation, the 3R-tau/4R-tau ratio correlates with the severity of tau aggregation, showing an increasing gradient from mildly affected to severely affected areas [24].

The tau-propagation theory may be compatible with the spread of 4R-tau in PSP and CBD but hardly explains the isoform shift observed in AD, because these isoforms are produced through alternative splicing of the *MAPT* gene. Moreover, events related to *MAPT*-splicing may not explain the isoform shift in tau aggregates in AD. Although several factors, including RNA-binding proteins [99], ribonucleoproteins [100], and splicing factors [101,102,103], have been reported to influence *MAPT* splicing, their roles remain mostly unknown in AD brains. It is known that mRNA levels of total or isoform-specific *MAPT* vary across studies and may not be in line with the isoform shift in tau in AD [104]. Indeed, a study reported that the 3R/4R mRNA ratio tended to be lower in advanced AD cases compared with normal controls or those with mild cognitive impairment [105]. While the intracellular growth of tau aggregates can be explained by seeding ability or post-translational events, the mechanisms underlying the 3R/4R isoform shift remain to be elucidated in AD.

## 4. Conclusions

According to the ‘amyloid cascade hypothesis’, Aβ deposition represents the upstream event that drives tau aggregation, leading to neuronal loss. However, neuropathological and experimental evidence indicates that the spaciotemporal relationship between tau and Aβ depositions is only partial and cannot fully explain the hierarchical pattern of tau propagation observed in AD.

Tau aggregation correlates more closely than Aβ deposition with neuronal loss, regional brain atrophy, and clinical severity. The stereotyped progression of tau pathology follows a proximity-dependent cortico-cortical pattern, beginning in the transentorhinal cortex and extending to the hippocampus and associative neocortex. This pattern does not strictly align with the connectivity of long-range memory circuits. In this respect, AD differs from the system-selective, poly-nodal degeneration observed in other 4R-tauopathies (PSP or CBD), ALS, and spinocerebellar degenerations. The exact determinants of the differential spreading manner of the tau aggregation (proximity-depending or systematic) between AD and 4R tauopathies remain largely elusive. In addition, our current understanding is limited for the dynamic isoform shift from 4R-tau-dominant to 3R-tau-dominant during NFT maturation in AD brains. Overall, tau pathology in AD is still enigmatic even more than 100 years after the first description of this disease. The pathogenesis of tau aggregation in AD cannot be explained simply by ‘amyloid cascade’ or ‘tau propagation along memory system’ alone, suggesting that there are missing pieces to fill the gaps between Aβ and tau pathologies or between AD and other tauopathies.

The emerging evidence indicates that while Aβ may facilitate tau aggregation, tau propagation in AD follows unique molecular and anatomical patterns that cannot be explained simply by the amyloid cascade. AD may thus represent a distinct tauopathy which is characterized by degeneration of local interactions among vulnerable neuronal populations, rather than global network degeneration, and the tauopathy powerfully leads disease progression of AD. Future research is necessary to identify the molecular triggers that initiate tau misfolding in early stages and to clarify how Aβ and tau pathologies interact to produce synaptic and neuronal loss.

The nature of tauopathy of AD might be related to the fact that treatment with anti-Aβ monoclonal antibodies (mAbs) demonstrates significant but partial improvement for clinical progression of AD despite successful reduction in Aβ deposits [106]. When considering that tau aggregation is facilitated by Aβ, the suppression of Aβ deposition seems to be a reasonable strategy to slow down the spreading of tau pathology during the early disease stages. In fact, studies revealed an improvement in tau-markers, including plasma p-tau181 and p-tau217 and tau-PET, after anti-Aβ treatment [107,108,109,110]. However, as discussed in Chapter 3, the tau-autonomous pathway may also exist in the spreading of tau pathology in AD, meaning that anti-Aβ mAbs does not completely halt the progression of tau aggregation. In this context, dual therapy targeting Aβ and tau [111] may become a feasible approach, although the essential questions, ‘what initiates tau aggregation?’ and ‘how does Aβ deposition facilitate tau aggregation?’, remain to be elucidated.

As evidenced by these results, neuropathology had and still has a pivotal role in research on AD and other neurodegenerative diseases in order for us to understand these interactions [112].

## Figures and Tables

**Figure 1 ijms-26-12088-f001:**
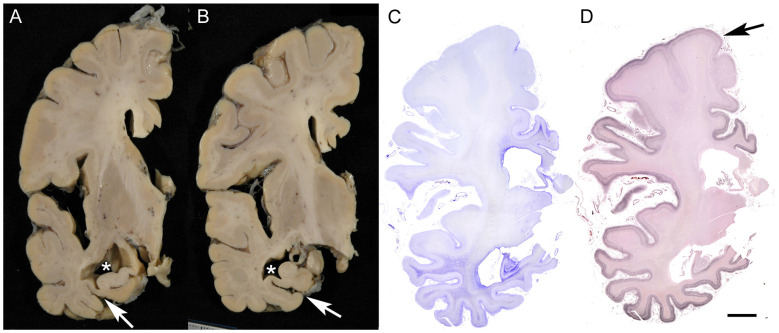
Macroscopic finding of AD brain. The panels show an AD case with a clinical duration of 8 years; the patient died at the age of 79. (**A**,**B**) The parahippocampal gyrus and hippocampus exhibit marked atrophy in association with dilatation of the lateral ventricle (*) and collateral sulcus (arrows). The atrophy is prominent in the posterior portion (**B**). (**C**,**D**) Pronounced astrogliosis is observed on Holzer staining (**C**), and strong argyrophilicity is evident on Gallyas silver impregnation (**D**) in the medial temporal areas and limbic cortices, showing decreasing gradient toward the frontal cortices with relative preservation of the precentral gyrus (arrow). Scale bar = 1 cm.

**Figure 2 ijms-26-12088-f002:**
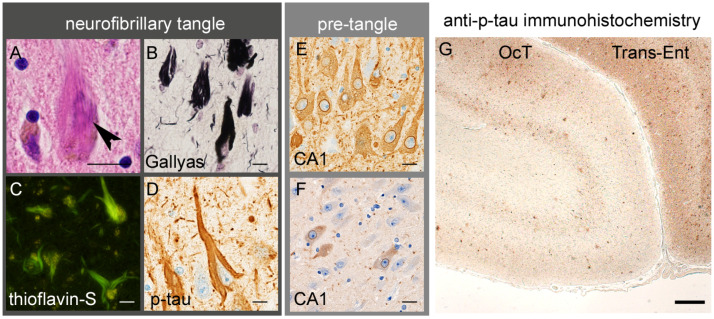
Tau aggregation in AD. (**A**–**D**) Neurofibrillary tangles (**A**, arrowhead) are argyrophilic and thioflavin-positive cytoplasmic inclusions, mainly composed of hyperphosphorylated tau (p-tau). (**E**) Pretangles are diffuse cytoplasmic tau aggregates, often showing strong immunoreactivity along the nuclear membrane. (**F**) Anti-tau-oligomer-specific immunohistochemistry recognizes pretangles. (**G**) In AD, dense tau aggregation in the neuronal cytoplasm and neurites results in a layer-specific pattern of immunostaining in the transentorhinal (Trans-Ent) cortex, one of the earliest sites of cerebral lesions, and to a lesser extent in the occipitotemporal (OCT) cortex. Scale bars: (**A**–**F**) 10 μm and (**G**) 100 μm. (**D**,**E**,**G**) Anti-p-tau (clone AT8, mouse monoclonal, 1:3000, Waltham, MA, USA) and (**F**) anti-tau oligomer (clone 2D6-2C6, rat monoclonal, 1:100 [20]) antibodies.

**Figure 3 ijms-26-12088-f003:**
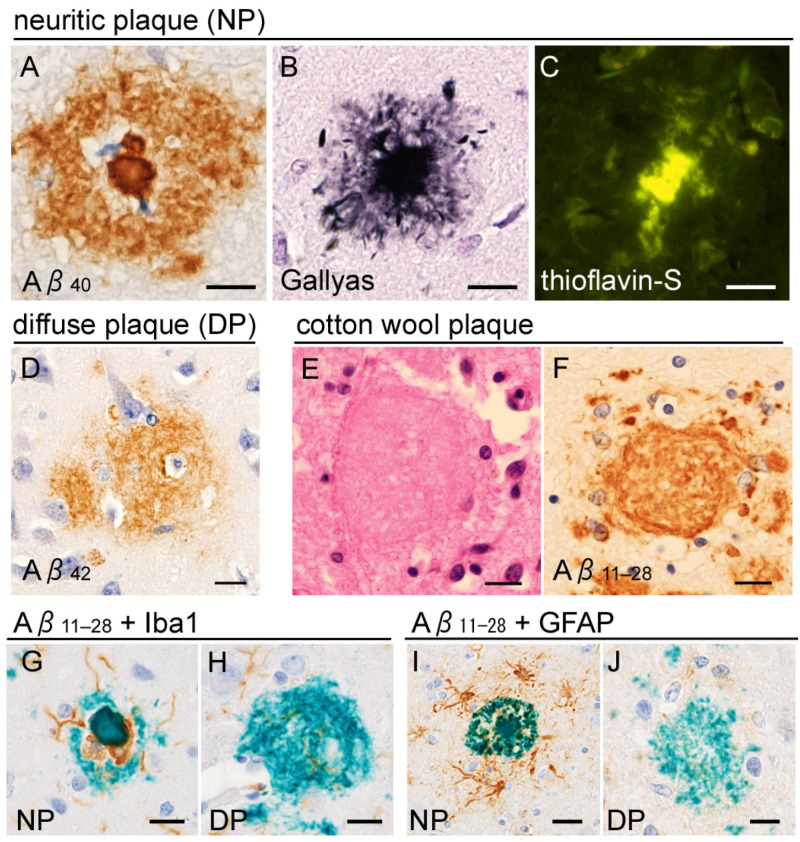
Senile plaques. (**A**–**D**) Neuritic plaques (NPs) are Aβ deposits within the neuropil, characterized by a dense core and peripheral halo (**A**). Anti-Aβ_40_ immunohistochemistry clearly labels the plaque core (**A**). The core is surrounded by strongly argyrophilic (**B**) and thioflavin-S-positive (**C**) crowns, composed of dystrophic neurites. (**D**) Diffuse plaques (DPs) are Aβ_42_-rich and lack the distinct core and crown structures. (**E**,**F**) Cotton wool plaques observed in a *PSEN1*-mutated case are shown (mutated site is not available in this case). They lack a core and crown but display a well-defined boundary, different from diffuse plaques. (**G**,**H**) Anti-Iba1 immunohistochemistry reveals microglial infiltration within the space between the core and halo (**G**), which is not observed in diffuse plaque (**H**). (**I**,**J**) Anti-glial fibrillary acidic protein (GFAP) immunohistochemistry shows that astroglia are prominent around the neuritic plaques, rather than inside. Scale bars (**A**–**J**) 10 μm. (**A**) Anti-Aβ_40_ (rabbit polyclonal, 1:500, IBL, Gunma, Japan), (**D**) anti-Aβ_42_ (rabbit polyclonal, 1:500, IBL), and (**F**–**J**) anti-Aβ_11–28_ (clone 12B2, mouse monoclonal, 1:1000, IBL) antibodies.

**Figure 4 ijms-26-12088-f004:**
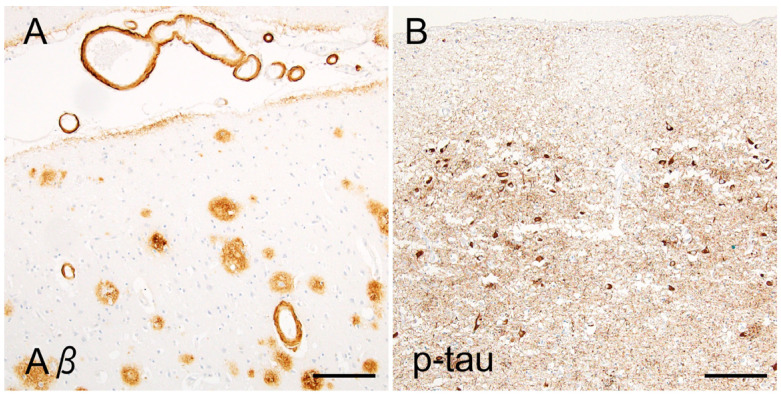
AD pathology in a Down syndrome case. The patient had Down syndrome (21 trisomy) and died at the age of 48. Down syndrome cases demonstrate not only senile plaques and cerebral amyloid angiopathy ((**A**), temporal neocortex, Aβ_11–28_) but also severe tau aggregates ((**B**), entorhinal cortex, AT8). Scale bars = 100 μm.

**Figure 5 ijms-26-12088-f005:**
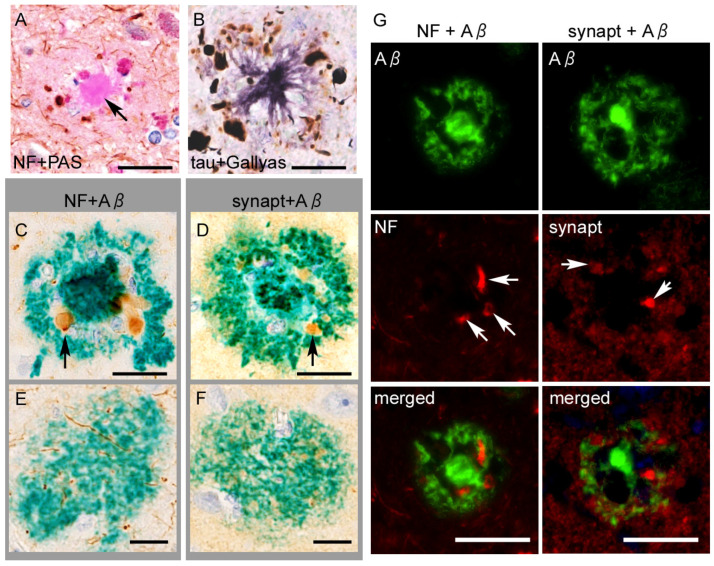
Neuroaxonal or tau epitopes on senile plaques. (**A**) Anti-neurofilament (NF) immunohistochemistry combined with PAS-staining shows swollen axons surrounding the plaque core. The core itself (arrow) is immunonegative. (**B**) Anti-p-tau (AT8) immunohistochemistry combined with silver impregnation reveals neuritic tau aggregates surrounding an amyloid core. (**C**–**F**) Neuritic plaques contain epitopes of NF and synaptophysin (synapt) ((**C**,**D**), arrows) around the plaque core, whereas diffuse plaques do not (**E**,**F**). (**G**) Double immunofluorescence shows dystrophic axons (arrows in the left column) or swollen terminals (arrows in the right column) are engulfed by Aβ_11–28_, without a clear co-localization. Scale bars = (**A**–**F**) 10 μm and (**G**) 20 μm.

**Figure 6 ijms-26-12088-f006:**
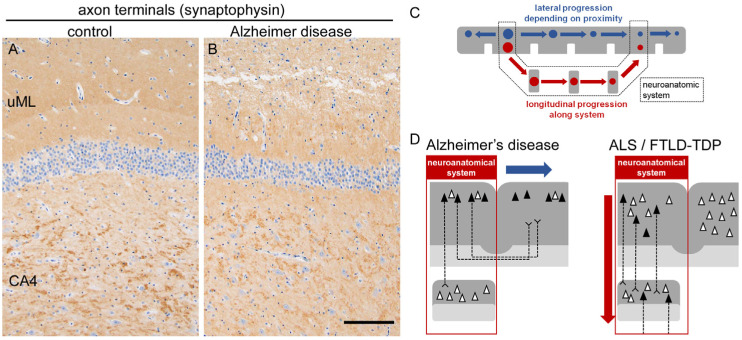
Synaptic loss and schematic of system degeneration. (**A**,**B**) In AD, axon terminals are depleted in the upper molecular layer (uML) of the hippocampal dentate gyrus, indicating loss of projection from the parahippocampal gyrus. Note the preservation of large synaptic boutons in CA4 projected from the dentate gyrus. Anti-synaptophysin immunohistochemistry. Scale bar = 100 μm. (**C**) Schematic illustration of lateral (blue) and longitudinal (red) modes of system degeneration. Lateral degeneration depends on spatial proximity and occurs in a mono-nodal manner when the system is considered. Longitudinal degeneration is poly-nodal and continuous along the system. Gray matter is depicted in gray, and circles indicate neuronal loss and protein aggregation at each node. (**D**) Simplified illustration of lateral spreading of AD-related tau pathology and longitudinal spreading of ALS/FTLD-related changes. Gray matter is shown as dense gray, white matter as light gray, and red frames indicate the neuroanatomical system. The filled triangles indicate neurons containing protein aggregates, whereas blank ones are not affected.

## Data Availability

No new data were created or analyzed in this study. Data sharing is not applicable to this article.

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
