# Peer review of "The Determinant of Tau Spreading in Alzheimer’s Disease: Dependent on Senile Plaque, Neural Circuits, or Spatial Proximity?"

_ijms, 2025, doi:10.3390/ijms262412088_

Round 1
Reviewer 1 Report
Comments and Suggestions for Authors
The manuscript titled “The determinant of tau spreading in Alzheimer’s disease: depending on senile plaque, neural circuits, or spatial proximity?” by Riku, Y.; et al. is a Review work where the authors outlined the most recent advances in the field of Alzheimer’s disease and the molecular factors that can trigger the onset and propagation of this devastating neurodegenerative malignancy. This review discusses about the remaining open question of the key pivotal factors that can lead to alzheimer’s disease pathogenesis. This is a topic of growing interest and the manuscript is generally well-written.
However, it exists some points that need to be addressed (please, see them below detailed point-by-point) to improve the scientific quality of the submitted manuscript paper before this article will be consider for its publication in the International Journal of Molecular Sciences.
1) Introduction. “Alzheimer’s disease (AD) encompasses a clinical and neuropathological entity (…) their pathogenic significance remained uncertain” (lines 44-54). Could the authors provide quantitative data insights according to the worldwide global burdens of Alzheimer’s disease and the related socieconomic cost? This will significantly aid the potential readers to better understand the significance of this conducted work.
2) “2.1. Neuropathology of Alzheimer’s disease” (lines 92-169). Here, the authors indicate that the Aβ deposits and neuritic plaques need to be assessed by thioflavin T fluorescent assays or silver impregantion (lines 97-99). It should not be neglected the use of nanoscale imaging tools [1] or positron emission tomography [2] to also visualize the Aβ deposits and plaques. This will strengthen the continuous need to develop ultrasensitive detection tools for the early prognosis of this disease.
[1] https://doi.org/10.1002/smsc.202500351
[2] https://doi.org/10.1038/s41380-025-03081-2
3) “In brief (…) Aβ42/Aβ40 (…)” (lines 172-173). The numbers should appear in subscript. This comment needs to be taken into account in the rest of the main manuscript body text.
4) Finally, comparing Aβ amyloid fibrils with neurotic plaques, what is more neurotoxic and the key driver for the onset of this disease?
5) “4. Conclusions” (lines 444-475). This section perfectly remarks the most relevant outcomes found by the authors in this field and the promising future prospectives. It may be remarkable to also briefly discuss about the potential future action lines to pursue the topic covered in this work.
Author Response
reviewer 1
1) Introduction. “Alzheimer’s disease (AD) encompasses a clinical and neuropathological entity (…) their pathogenic significance remained uncertain” (lines 44-54). Could the authors provide quantitative data insights according to the worldwide global burdens of Alzheimer’s disease and the related socieconomic cost? This will significantly aid the potential readers to better understand the significance of this conducted work.
Answer: We appreciate this suggestion. In the revised version, some writings about social impacts derived from AD has been added to the introduction (the lines 44-51).
2) “2.1. Neuropathology of Alzheimer’s disease” (lines 92-169). Here, the authors indicate that the Aβ deposits and neuritic plaques need to be assessed by thioflavin T fluorescent assays or silver impregantion (lines 97-99). It should not be neglected the use of nanoscale imaging tools [1] or positron emission tomography [2] to also visualize the Aβ deposits and plaques. This will strengthen the continuous need to develop ultrasensitive detection tools for the early prognosis of this disease.
[1] https://doi.org/10.1002/smsc.202500351
[2] https://doi.org/10.1038/s41380-025-03081-2
Answer: We thank for the suggestion and good literatures. The lines 65-69 of revised paper address biomarkers for Aβ deposition and phosphorylated tau.
3) “In brief (…) Aβ42/Aβ40 (…)” (lines 172-173). The numbers should appear in subscript. This comment needs to be taken into account in the rest of the main manuscript body text.
Answer: We have changed all writings of Aβ species to be in subscript styles.
4) Finally, comparing Aβ amyloid fibrils with neurotic plaques, what is more neurotoxic and the key driver for the onset of this disease?
Answer: It is difficult to directly confirm which is more neurotoxic for AD patients, hence we did not clarify this point in the paper. At least, as discussed in the chapters 2 and 3, neuritic plaques are more closely associated with tau aggregation, synaptic alteration, neuroinflammation, and severe cognitive impairment than diffuse plaques. This is, however, correlation but not causality. It is not conclusive in AD patients whether Aβ fibrils/protofibrils in diffuse plaques remain to be benign or not. We guess that future autopsy studies on AD patients who have undergone anti- Aβ mAbs in a large sample size will be informative of this issue because early AD patients (who may be in diffuse plaque-dominant state) are mainly targeted by this therapy.
5) “4. Conclusions” (lines 444-475). This section perfectly remarks the most relevant outcomes found by the authors in this field and the promising future prospectives. It may be remarkable to also briefly discuss about the potential future action lines to pursue the topic covered in this work.
Answer: We appreciate this suggestion. We consider that the understanding neuropathology (particularly of tau) of AD provides important insights for interpretation of anti-Aβ/tau mAb therapies. For example, why does anti-Aβ therapy demonstrate significant but partial effects for clinical AD progression? Should we address dual therapy toward both Aβ and tau? If the dual therapy is technically possible, can it be preventive of AD? We think, a viewpoint of ‘tauopathy’ is necessary to address these questions. Some discussions about this point have been added into the conclusion section (the lines 482-492).
Reviewer 2 Report
Comments and Suggestions for Authors
This is a narrative review of amyloidopathy and especially tauopathy in Alzheimer’s disease (AD), their possible mechanisms of interplay and controversies or unresolved issues.
This review is very well written and educative and follows the typical methodology for narrative reviews. References are appropriate. All figures are significant, but especially Fig. 6 is very illustrative and providing explanation of the mode of tau propagation in AD.
One point: What is the significance of all these findings in every day clinical practice and especially in AD therapeutics? Monoclonal anti-Aβ antibodies have been recently introduced in the treatment of AD and, indeed, they show not only reduction of Aβ burden, but also some degree of reduction of p-tau in the CSF. It is very well described in the manuscript that it is not Aβ, but tau that correlates with clinical severity. Is this the reason why anti-Aβ antibodies show a mild-moderate therapeutic efficacy? Would the combination of anti-Αβ and anti-tau treatments expected to be more effective? Please add a few comments on that, in the discussion section.
Author Response
One point: What is the significance of all these findings in every day clinical practice and especially in AD therapeutics? Monoclonal anti-Aβ antibodies have been recently introduced in the treatment of AD and, indeed, they show not only reduction of Aβ burden, but also some degree of reduction of p-tau in the CSF. It is very well described in the manuscript that it is not Aβ, but tau that correlates with clinical severity. Is this the reason why anti-Aβ antibodies show a mild-moderate therapeutic efficacy? Would the combination of anti-Αβ and anti-tau treatments expected to be more effective? Please add a few comments on that, in the discussion section.
Answer: We deeply appreciate the reviewer’s insights from therapeutic aspects. The biggest question for anti- Aβ therapy is significant but partial effect of it for clinical AD progression, despite of remarkable reduction of Aβ deposition. We speculate that existence of tau-autonomous manner in AD-related tauopathy may be a factor for it. Hence, anti-Aβ therapy may slow down the progression of tau pathology but not completely halt it. Dual therapy toward Aβ and tau aggregations might be a feasible way, but the essential questions about ‘initiator of tau aggregation (Aβ may be facilitator but not initiator of tau pathology)’ and ‘interaction between Aβ and tau (how does Aβ facilitate tau aggregation?)’ should be more intensely approached. Some discussions about this point have been added into the conclusion section (the lines 482-492).
Round 2
Reviewer 2 Report
Comments and Suggestions for Authors
The authors modified the text according to suggestions and the manuscript has been sufficiently improved to warrant publication.